**Subject Category:**
Biology (whole organism)

biomedical engineering/biomechanics

joint angle, paralympics, locomotion, prosthetics, performance analysis, adaptation

**Author for correspondence:**
Johannes Funken
e-mail: j.funken@dshs-koeln.de

# Long jumpers with and without a transtibial amputation have different three-dimensional centre of mass and joint take-off step kinematics

Johannes Funken[1], Steffen Willwacher[1,2], Kai Heinrich[1], Ralf Müller[1], Hiroaki Hobara[3], Alena M. Grabowski[4,5] and Wolfgang Potthast[1,6]

[1]Institute of Biomechanics and Orthopaedics, German Sport University Cologne, Am Sportpark Müngersdorf 6, 50933 Cologne, Germany
[2]Institute of Functional Diagnostics, Im Mediapark 2, 50670 Cologne, Germany
[3]Artificial Intelligence Research Center, National Institute of Advanced Industrial Science and Technology, 2-3-26, Aomi, Koto-ku, Tokyo 135-0064, Japan
[4]Integrative Physiology Department, University of Colorado, 354 UCB, Boulder, CO 80309-0354, USA
[5]Department of Veterans Affairs, Eastern Colorado Healthcare System, Denver, CO, USA
[6]ARCUS Clinics, Rastatter Strasse 17–19, 75179 Pforzheim, Germany

JF, 0000-0002-0093-8686; SW, 0000-0002-1303-3165; KH, 0000-0001-9417-9619; HH, 0000-0003-4710-9962; AMG, 0000-0002-4432-618X; WP, 0000-0001-9135-3802

Long jumpers with below the knee amputation (BKA) have achieved remarkable performances, yet the underlying biomechanics resulting in these jump distances are unknown. We measured three-dimensional motion and used multi-segment modelling to quantify and compare the centre of mass (COM) and joint kinematics of three long jumpers with BKA and seven non-amputee long jumpers during the take-off step of the long jump. Despite having the same jump distances, athletes with BKA, who used their affected leg for the take-off step, had lower sagittal plane hip and knee joint range of motion and positioned their affected leg more laterally relative to the COM compared to non-amputee athletes. Athletes with BKA had a longer compression phase and greater downward movement of their COM, suggesting that their affected leg (lever) was less rigid compared to the biological leg of non-amputees. Thus, athletes with BKA used a different kinematic

mechanism to redirect horizontal to vertical velocity compared to non-amputee athletes. The specific movement patterns of athletes with BKA during the take-off step were constrained by the mechanical properties of the prosthesis. These results provide a basis for coaches and athletes to develop training protocols that improve performance and inform the design of future prostheses.

## 1. Background

Humans are capable of adapting the way they move to accomplish a wide range of bipedal movement tasks [1–4]. These adaptations also include those made by athletes with leg amputations using sport-specific prostheses to run, sprint and jump [5–7]. Long jumpers with a unilateral below the knee amputation (BKA) who use a carbon fibre running-specific prosthesis (RSP), for example, have achieved remarkable jump distances [7]. The current long jump world record for male athletes with BKA (8.48 m) would have resulted in a gold medal in the previous three Olympic Games and at least a bronze medal in all Olympic Games ever [8], where the world record for non-amputees is 8.95 m. However, the underlying biomechanics of such elite long jumpers with BKA are not known but are important for understanding how these athletes adapt the way that they move to perform the long jump and may also be used by coaches and athletes to enhance training and performance.

A previous study reported that male athletes with BKA ($n = 8$) showed the same basic long jump technique as male non-amputee long jumpers [9]. However, this study measured data from athletes with BKA competing in the 1998 World Disabled Athletics Championship where the average actual jump distance was 6.00 m, and only one athlete used his affected leg as his take-off leg. During the long jump final of the 2002 World Disabled Athletics Championships, a new world record (6.79 m) was achieved by a male athlete with unilateral BKA using his unaffected leg as his take-off leg [10]. There were no significant differences in jump distance during the 2004 Paralympic Games due to the take-off leg, but take-off technique differed between male athletes with BKA who used their affected leg as their take-off leg ($n = 5$, $6.04 \pm 0.66$ m) compared to those who used their unaffected leg ($n = 5$, $5.22 \pm 0.73$ m) as their take-off leg [11]. Athletes who used their affected leg for the take-off step had less sagittal plane hip joint range of motion (ROM) and a stiffer knee joint compared to athletes who used their unaffected leg for the take-off step [11]. Since 2002, the long jump world record of male athletes with BKA has improved by about 1.7 m, and today almost all elite long jumpers with BKA use their affected leg as their take-off leg. In line with Nolan *et al.* [11], we found that the underlying centre of mass (COM) [7] and joint kinetics [7,12] were fundamentally different throughout the take-off step between three male long jumpers with BKA who used their affected leg as their take-off leg compared to male non-amputee long jumpers. However, specific take-off parameters, which directly determine jump distance, such as take-off angle, COM height and velocity, were similar at the end of the take-off step for the best long jumper with BKA (personal record (PR) at the time of the study: 8.40 m) and the best non-amputee long jumper (PR: 8.52 m) [7]. In order to identify the underlying reasons for the kinetic differences during the take-off step [7,12], it is necessary to determine COM and joint kinematics not only for the end of the take-off step but throughout the entire stance phase. Furthermore, it is important to understand the adaptations and constraints that are potentially induced by using RSPs. Numerous studies have determined the two-dimensional kinematics [13–16] and the three-dimensional kinematics [17–20] of non-amputee athletes during the long jump take-off step. To our knowledge, no previous studies have determined the three-dimensional kinematics of long jumpers with BKA who use RSPs and compete on a recent performance level. Since the use of an RSP results in different biomechanics for athletes with BKA during running and sprinting compared to non-amputees [21–23], the sagittal, as well as frontal and transverse plane kinematics during the long jump take-off step, may also differ between athletes with and without a BKA. Determining the three-dimensional kinematics of athletes with BKA using an RSP and their affected leg as their take-off leg during the long jump compared to non-amputees will provide information that can be used to improve training techniques and prosthetic design. A comprehensive three-dimensional analysis of the biomechanical movement patterns elicited by athletes with BKA will, furthermore, generate valuable insight about the long jump and jumping locomotion in general.

Therefore, the aim of the present study was to quantify the three-dimensional long jump take-off step COM and joint kinematics of athletes with BKA and compare them to those elicited by non-amputee athletes.

**Table 1.** PR for the long jump and anthropometrics for athletes with BKA and non-amputee athletes (non-AMP).

| group | PR (m) | mass[a] (kg) | height (m) | age (years) |
|---|---|---|---|---|
| BKA ($n = 3$) | 7.43 ± 0.99 | 78.7 ± 9.8 | 1.83 ± 0.04 | 26.0 ± 1.7 |
| non-AMP ($n = 7$) | 7.65 ± 0.65 | 80.1 ± 6.2 | 1.82 ± 0.07 | 24.6 ± 2.5 |

[a]Body mass of the athletes with BKA includes the prosthesis.

# 2. Methods

## 2.1. Participants and study design

Ten male athletes gave voluntary informed consent to participate in the study and were divided into two groups (table 1). The first group (BKA) comprised three athletes with a BKA on their right side and the second group (non-AMP) included seven non-amputee athletes. All athletes with BKA used the same type of RSP (Cheetah Xtreme; Össur, Reykjavik, Iceland) with their individual alignment. Data collection was conducted at the German Sport University Cologne (GSU, two athletes with and six without BKA) and the Japan Institute of Sport Sciences (JISS, one athlete with and one without BKA). The study design was in line with the declaration of Helsinki and was approved by the GSU ethical committee board (approval number: 040/2016).

## 2.2. Data collection

We measured each athlete's lower and upper body segment lengths and circumferences with a tape measure and an anthropometer. Body height was measured while athletes stood upright with both legs loaded. We attached retroreflective markers to anatomic reference points on the athlete's body and prosthesis using double-sided tape [7]. Before testing, each athlete performed an individual competition specific warm-up and completed practice long jumps to get used to the measurement set-up. The athletes were asked to perform three to six maximum distance long jumps with their individual, competition specific maximum run-up. A three-dimensional motion capture system (Vicon, Oxford, UK) operating at 250 Hz (GSU) or 500 Hz (JISS), respectively, was used to collect kinematic data during the take-off step. One force plate mounted flush with the floor (1000 Hz, Kistler Instrumente Corporation, Winterthur, Switzerland) captured kinetic data of the take-off step. Three high-speed video cameras (100 Hz, Basler, Ahrensburg, Germany) with different points of view were used for capturing qualitative videos and to ensure valid force plate strikes between touchdown (TD) and toe-off (TO). All athletes with BKA used their affected right leg as their take-off leg. Four non-amputee athletes used their right leg and three used their left leg for the take-off step.

## 2.3. Data analysis

Kinematic and kinetic data were both filtered with the same cut-off frequency [24] of 50 Hz using a fourth-order recursive Butterworth filter [25]. Ground contact (from first (TD) to last (TO) frame with the foot on the ground) was determined using the vertical component of the ground reaction force with a threshold of 10 N [12]. The compression phase was defined as the time interval from TD to maximum knee flexion (MKF) and the extension phase was defined as the time interval from MKF to TO [9,15]. A mathematical rigid full body model (Dynamicus, Alaska, Institute of Mechatronics, Chemnitz, Germany) consisting of 16 main segments was used for kinematic calculations for the non-amputee athletes. The segments comprised: head, trunk, as well as left and right: upper arm, lower arm, hand, thigh, shank, rear foot and forefoot. The trunk consisted of several subsegments. For athletes with BKA, the same model was used, but modified by replacing the foot and lower shank of the right leg with a prosthesis (figure 1). To detect frontal, transverse and sagittal plane motion, we chose to reconstruct the prosthesis as a two-segment rigid body connected with a ball joint. The prosthetic 'ankle joint' refers to the point on the prosthesis with the highest curvature [21,26], which coincides with its most posterior point. The 'ankle joint' axis was determined from two markers attached to the medial and lateral edge of the RSP [7,26]. The anthropometric data, as well as mass and dimensions of the prosthesis, were used to adapt each model to the individual body dimensions

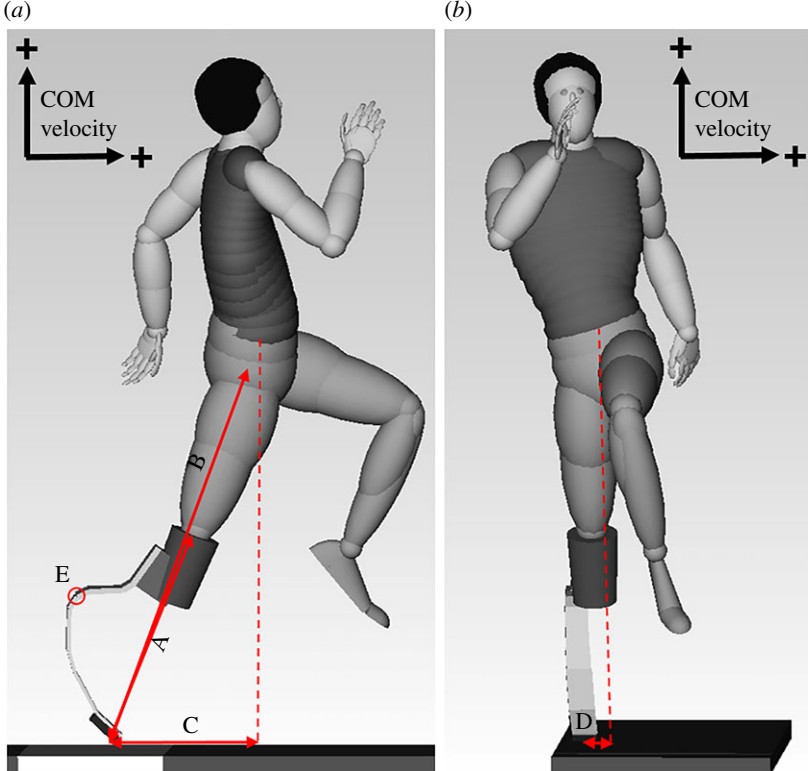

**Figure 1.** Model sagittal (*a*) and frontal (*b*) plane views including COM velocity orientation. Red arrows indicate (A) the lower leg length, (B) the whole leg length, (C) the antero-posterior distance of the COM to the centre of pressure (COP) and (D) the medio-lateral distance of the COM to the COP. The red circle indicates the position of the prosthetic ankle joint (E).

and RSP of each athlete. One reference trial was captured with athletes in a static erect position to define the segmental coordinate systems.

COM displacement during the take-off step was calculated in the global laboratory coordinate system for all three axes as the distance of the COM coordinate to the respective COM coordinate at the instant of TD. COM velocity was calculated by mathematical differentiation of the COM position throughout the entire stance phase. To account for the take-off foot laterality, COM velocity and displacement are expressed in the global coordinate system with respect to the take-off foot. A velocity described as 'lateral', therefore, represents COM velocity in the lateral direction relative to the take-off foot (e.g. to the right for an athlete using the right leg for the take-off step).

In order to identify differences in foot position, the antero-posterior and medio-lateral distances from the COM to the centre of pressure (COP) of the take-off foot were determined. Additionally, lengths of the whole leg and lower leg were calculated as the linear distance from the COP to the hip joint centre or the linear distance from the COP to the knee joint centre, respectively (figure 1). All joint angles represent the rotation of the coordinate system attached to the distal segment with respect to the coordinate system attached to the proximal segment of the respective joint. To ensure a reasonable comparison of jump distances between the two groups, the influence of landing technique was eliminated by calculating the theoretical jump distance as in our previous publication [7]. The best jump with respect to the theoretical jump distance of each athlete was used for analysis of the take-off step. Since we analysed the same jumps as in our previous publication [7], we also adopted the previously calculated theoretical jump distances. Further details on the mathematical model (e.g. joint centre definition) and calculation of jump distance are in Willwacher *et al.* [7].

## 2.4. Statistics

Due to the small sample size, a non-parametric Wilcoxon ranked-sum test was used to identify differences between the two groups and the margin of error was set to 10% [7]. We assumed that the null hypothesis was indicated as no difference between groups. We also present the percentage difference for athletes with BKA relative to the non-amputee athletes.

# 3. Results

The average long jump distance for athletes with BKA (7.26 ± 0.77 m) was not different compared to non-amputee athletes (7.27 ± 0.45 m) ($p = 1.000$). Athletes with BKA reached MKF in their take-off leg later during the take-off step compared to the non-amputee athletes (non-AMP: 43.8 ± 4.9%, BKA: 52.7 ± 2.5%, $p = 0.017$).

## 3.1. COM kinematics

COM displacement in the antero-posterior (jumping) direction was not different between groups during the take-off step—neither with regard to total displacement ($p = 1.000$) nor the distance of the COM relative to the COP at TO ($p = 0.517$) (table 2 and figures 2 and 3). At the instant of TD, the COM of the non-amputee athletes was approximately 10 cm closer to the COP compared to athletes with BKA ($p = 0.067$). At MKF, both groups had their COM posterior relative to their COP, but athletes with BKA had their COM approximately 13 cm closer to their COP compared to non-amputees ($p = 0.017$).

Medio-lateral COM displacement during stance was different between groups ($p = 0.033$). Non-amputees had a lateral (relative to their take-off foot) COM displacement (2.6 cm), while athletes with BKA did not have relevant medio-lateral movement during the take-off step (figure 2 and table 2). The COM position was approximately 5 cm medial to the COP in athletes with BKA during most of the stance phase, while for the non-amputee athletes, the COM was above the COP for most of stance, and was 1.4 cm lateral to the COP at TO (table 2 and figure 3).

In athletes with BKA, the vertical COM position was below their vertical COM position at TD for a longer relative duration of ground contact compared with non-amputee athletes (non-AMP: 17.2 ± 3.9%, BKA: 42.2 ± 5.8%, $p = 0.017$).

The downward vertical COM displacement in the athletes with BKA was 0.9 cm greater ($p = 0.017$), but total vertical COM displacement was 18.7% lower ($p = 0.017$) compared to non-amputee athletes (table 2 and figure 2).

Athletes with BKA had 7.5% slower horizontal velocity at TD ($p = 0.067$), lost 46.0% less horizontal velocity during stance ($p = 0.017$) and their horizontal velocity at TO was not different ($p = 0.667$) compared to non-amputee athletes (figure 2 and table 2). Vertical velocity at TD and TO was not different between groups ($p = 0.383$, $p = 1.000$). Medio-lateral COM velocity was close to zero at TD for both groups. During ground contact, medio-lateral COM velocity was near constant for athletes with BKA, but increased by $0.35 \, \mathrm{m\,s^{-1}}$ in the lateral direction for non-amputee athletes and was different at TO between groups ($p = 0.033$).

During the take-off step, total change in whole leg length (non-AMP: 18.7 ± 5.6 cm, BKA: 17.9 ± 1.9 cm) and in lower leg length (non-AMP: 16.9 ± 6.0 cm, BKA: 16.9 ± 2.4 cm) of the take-off leg were not different in athletes with BKA compared to non-amputee athletes ($p = 0.833$ and 0.383, respectively) (figure 3 and table 2).

During the compression phase of the take-off step (TD–MKF), shortening of the whole leg (non-AMP: 6.5 ± 4.7 cm, BKA: 16.8 ± 3.2 cm) and shortening of the lower leg (non-AMP: 5.4 ± 3.4 cm, BKA: 16.0 ± 3.6 cm) were 159.8% and 194.3% greater, respectively, in athletes with BKA compared to non-amputee athletes (both: $p = 0.017$). During the extension phase of the take-off step (MKF–TO), lengthening of the whole leg (non-AMP: 18.1 ± 5.7 cm, BKA: 17.0 ± 1.6 cm) and lengthening of the lower leg (non-AMP: 15.7 ± 6.2 cm, BKA: 15.8 ± 1.9 cm) were not different between both groups ($p = 1.000$ and 0.267, respectively).

## 3.2. Joint angles

Athletes with BKA had smaller sagittal plane ROM for all joints compared to non-amputee athletes (all $p = 0.017$) (figure 4). The passive-elastic RSP used by athletes with BKA displayed peak dorsiflexion of 27.9 ± 3.7° and only minimal plantarflexion during stance, while the biological ankle joint of non-amputee athletes displayed both dorsiflexion and plantarflexion with peak values of 17.6 ± 4.1° and 31.5 ± 5.5, respectively. The peak knee and hip flexion, but not extension, were lower in athletes with BKA compared to non-amputee athletes (both $p = 0.017$) (figure 4; electronic supplementary material, table S1).

Non-amputee athletes had discontinuous hip extension with a flexion angle at TD of 24.8 ± 4.8° and peak hip flexion of 34.4 ± 5.1° at approximately 25% of ground contact during the take-off step. Their hip

**Table 2.** COM kinematics. The mean values with standard deviations (s.d.), at the instances of TD, MKF and TO for athletes with BKA ($n = 3$) and non-amputee athletes (non-AMP, $n = 7$) during the take-off step of the long jump. Bold values indicate significant differences between groups.

| measures | athletes with BKA | | | non-amputee athletes | | |
|---|---|---|---|---|---|---|
| | TD | MKF | TO | TD | MKF | TO |
| COM displacement during stance (cm) | | | | | | |
| anterior | 0 | 54.2 (5.8) | 100.9 (6.2) | 0 | 47.2 (7.8) | 102.8 (8.9) |
| medio(+)/lateral(−) | 0 | 0.3 (0.7) | **0.2 (1.3)** | 0 | −0.5 (0.7) | **−2.6 (1.5)** |
| vertical | 0 | 1.8 (1.6) | **16.7 (1.6)** | 0 | 4.0 (1.9) | **21.8 (3.1)** |
| distances (cm) | | | | | | |
| anterior (+)/posterior (−) COM to COP | **−63.2 (2.1)** | **−7.3 (4.7)** | 36.7 (4.2) | **−53.2 (8.8)** | **−20.5 (5.8)** | 43.6 (14.5) |
| medio(+)/lateral(−) COM to COP | 8.0 (4.8) | **4.9 (1.1)** | 3.4 (2.0) | 0.6 (11.3) | **−0.9 (2.7)** | −1.4 (7.7) |
| whole leg length | **110.0 (1.5)** | 93.1 (3.0) | 110.1 (1.5) | **99.5 (4.3)** | 93.0 (4.1) | 111.1 (7.9) |
| lower leg length | **65.6 (2.5)** | 49.6 (4.6) | 65.4 (2.7) | **57.6 (3.1)** | 52.2 (2.3) | 67.8 (7.3) |
| COM velocity (m s⁻¹) | | | | | | |
| anterior | **8.74 (0.59)** | 8.05 (0.52) | 8.13 (0.52) | **9.46 (0.32)** | 8.52 (0.30) | 8.32 (0.35) |
| medio(+)/lateral(−) | 0.03 (0.13) | 0.0 (0.11) | **−0.04 (0.13)** | −0.03 (0.11) | −0.21 (0.20) | **−0.38 (0.21)** |
| vertical | −0.51 (0.17) | 1.68 (0.10) | 2.86 (0.28) | −0.41 (0.11) | 1.84 (0.44) | 3.00 (0.28) |

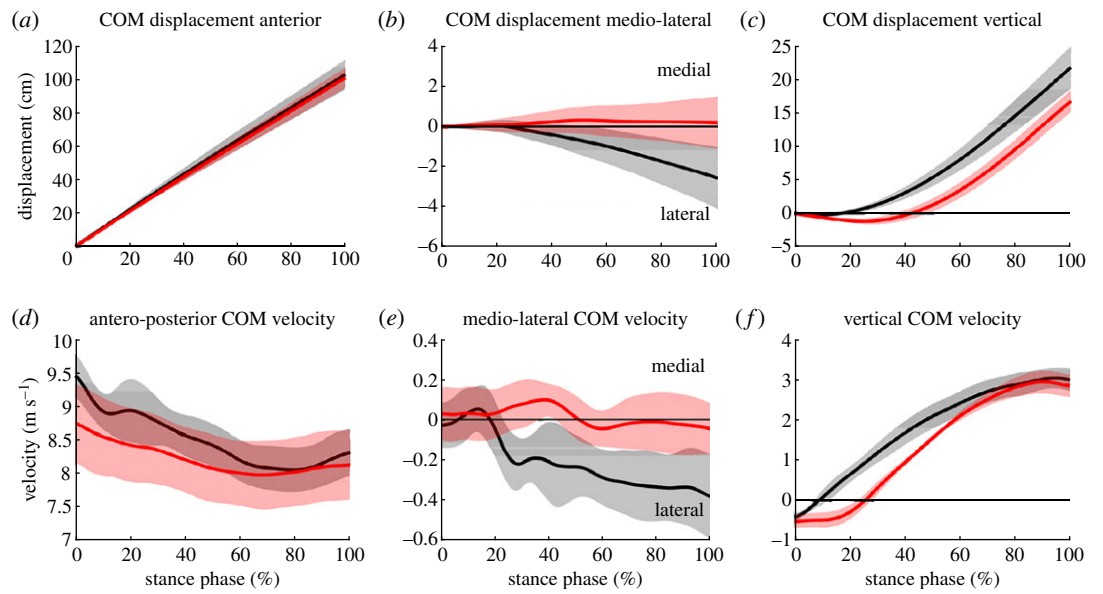

**Figure 2.** COM displacement ($a$–$c$) and velocity ($d$–$f$) during the take-off step stance phase for athletes with BKA ($n = 3$, red) and non-amputee athletes ($n = 7$, black) in the antero-posterior ($a,d$), medio-lateral ($b,e$) and vertical ($c,f$) directions.

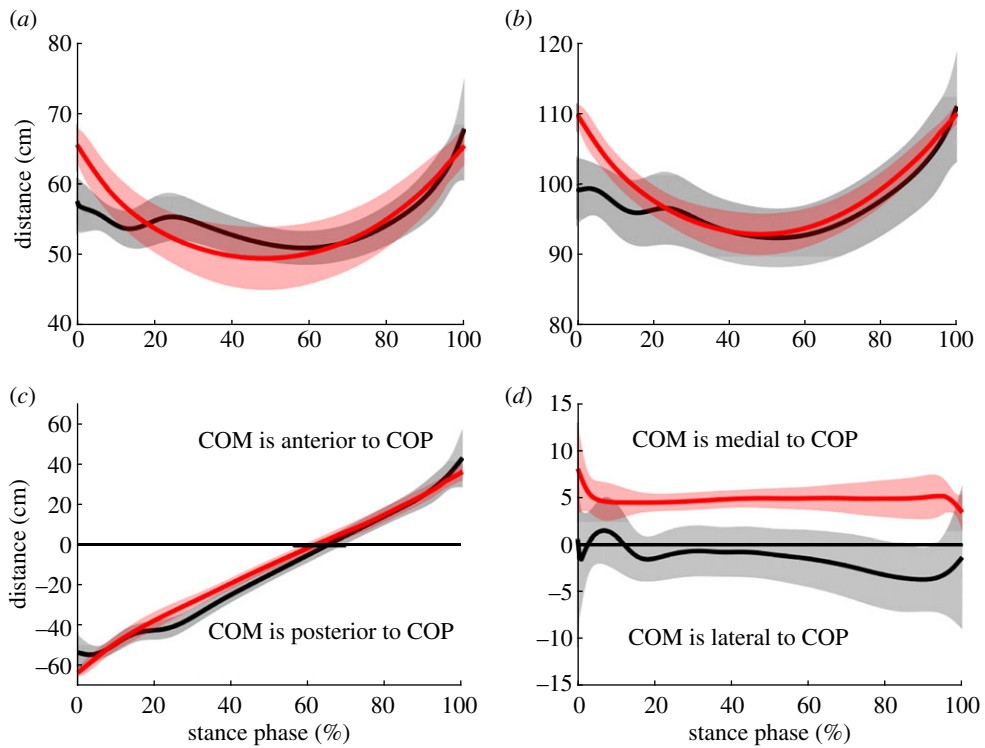

**Figure 3.** Lower leg length ($a$) and whole leg length ($b$) during the stance phase of the take-off step. Linear distance from the COM to the COP in the antero-posterior ($c$) and medio-lateral ($d$) directions during the stance phase of the take-off step. Athletes with BKA ($n = 3$) are indicated in red and non-amputee athletes ($n = 7$) are indicated in black.

extension angle at TO was $25.7 \pm 3.4°$. The sagittal plane hip angle of athletes with BKA displayed a near continuous extension and was not different at TD or TO compared to non-amputee athletes (both $p = 0.117$). However, at the instant of MKF, hip flexion was 72.6% lower in athletes with BKA compared to non-amputee athletes ($p = 0.017$) (table 3). In both groups, the hip joint angle switched from flexion to extension at about 70% of stance (BKA $68.2 \pm 7.5\%$; non-AMP $70.4 \pm 4.4\%$). The knee flexion angle was 38.7% lower at TD ($p = 0.033$) and 40.3% lower at MKF ($p = 0.017$) but not different at TO ($p = 0.517$) for athletes with BKA compared to non-amputee long jumpers.

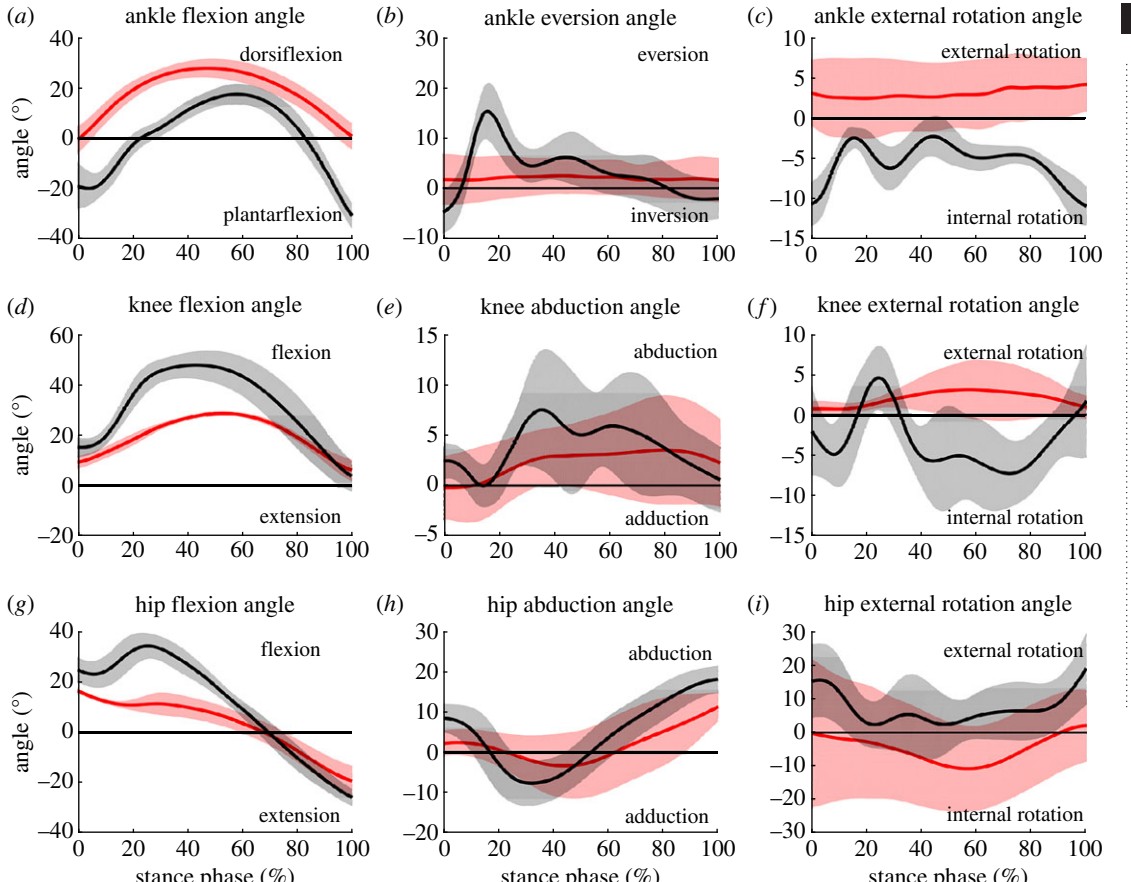

**Figure 4.** Joint angles during the stance phase of the take-off step for athletes with BKA ($n = 3$, red) and non-amputee athletes ($n = 7$, black) for the ankle ($a$–$c$), knee ($d$–$f$) and hip ($g$–$i$) in the sagittal ($a,d,g$), frontal ($b,e,h$) and transverse ($c,f,i$) planes.

Frontal and transverse plane prosthetic ankle angles of athletes with BKA were near constant throughout ground contact of the take-off step, whereas the ankle joint of non-amputee athletes displayed frontal and transverse plane ROMs of $20.6 \pm 6.9°$ and $10.6 \pm 3.7°$, respectively (figure 4; electronic supplementary material, table S1). At the instant of TD, frontal and transverse plane joint angles were not different between groups in the knee ($p = 0.183$, $p = 0.517$) and hip ($p = 0.117$, $p = 0.267$) (figure 4 and table 3). At MKF, the knee joint of the non-amputee athletes was internally rotated, whereas the knee joint of the athletes with BKA was externally rotated. At TO, athletes with BKA had 38.0% lower hip abduction and 89.1% lower hip external rotation compared to non-amputee athletes (both $p = 0.067$).

## 4. Discussion

The purpose of the current study was to quantify three-dimensional long jump take-off step COM and joint kinematics of athletes with BKA and compare those to non-amputee athletes. Long jumpers with BKA, who use their affected leg as their take-off leg, have different COM and joint kinematics throughout the stance phase of the take-off step compared to non-amputee athletes.

### 4.1. Sagittal plane COM kinematics

During the take-off step, non-amputee long jumpers use a mechanical mechanism called pivoting, where they rotate their COM about their foot to redirect a portion of the horizontal velocity into vertical velocity [15,27]. However, the pivot may be more effective if the athlete has greater eccentric strength providing a greater ability to resist knee flexion [14,15,27]. Resisting knee flexion would allow non-amputee athletes to keep their leg stiff, providing a stable lever. Athletes with BKA, even though they resist knee flexion more than non-amputees (figure 4 and table 3), had greater shortening of the whole leg and lower leg

**Table 3.** Joint angles. Joint angles as mean with standard deviations (s.d.), at the instances of TD, MKF and TO for the athletes with BKA ($n = 3$) and the non-amputee athletes (non-AMP, $n = 7$) during the take-off step of the long jump. (Dorsi)flexion, eversion/abduction and external rotation are indicated by positive values, whereas plantarflexion/extension, inversion/adduction and internal rotation are indicated by negative values. Bold values indicate significant differences between groups.

| joint angles (°) | athletes with BKA | | | non-amputee athletes | | |
|---|---|---|---|---|---|---|
| | TD | MKF | TO | TD | MKF | TO |
| ankle[a] | | | | | | |
| dorsiflexion (+) | −0.5 (5.1) | **27.5 (4.2)** | **0.7 (5.1)** | **−18.8 (9.1)** | **13.2 (3.9)** | **−30.8 (5.2)** |
| eversion (+) | **1.7 (5.0)** | 2.2 (3.4) | 1.6 (4.3) | **−4.8 (4.2)** | 5.7 (5.1) | −2.1 (4.0) |
| external rotation (+) | **3.2 (4.1)** | **3.0 (4.0)** | **3.3 (4.2)** | **−10.6 (2.7)** | **−2.9 (3.2)** | **−10.9 (2.4)** |
| knee | | | | | | |
| flexion (+) | **9.4 (2.4)** | **28.7 (0.5)** | 6.2 (4.3) | **15.3 (3.9)** | **48.2 (5.3)** | 3.8 (5.9) |
| abduction (+) | −0.2 (3.2) | 3.2 (3.4) | 2.3 (4.4) | 2.4 (1.8) | 5.2 (6.3) | 0.6 (3.3) |
| external rotation (+) | 0.8 (1.0) | **3.2 (3.3)** | 1.0 (1.4) | −2.0 (5.4) | **−4.1 (6.2)** | 1.8 (7.1) |
| hip | | | | | | |
| flexion (+) | 16.6 (0.8) | **6.5 (2.9)** | −19.1 (5.9) | 24.8 (4.8) | **23.6 (3.9)** | −25.7 (3.4) |
| abduction (+) | 2.3 (3.1) | −2.5 (8.2) | **11.3 (3.7)** | 8.4 (3.6) | −4.2 (4.9) | **18.1 (3.3)** |
| external rotation (+) | −0.2 (22.1) | **−10.6 (12.9)** | **2.1 (10.7)** | 15.3 (11.1) | 2.4 (11.3) | **19.2 (10.8)** |

[a]Ankle in the BKA group refers to the point of the prosthesis' greatest curvature defined as the prosthetic ankle joint.

during the compression phase of the take-off step (figure 3 and table 2), indicating that their leg is not as rigid as it is for non-amputee athletes. Because leg stiffness is dictated by RSP stiffness during running [28], the use of a stiffer prosthesis by an athlete with BKA could increase leg stiffness during the long jump take-off step. However, a stiffer prosthesis might impair the elastic energy storage and return within the leg and result in higher impact forces leading to increased knee flexion. Therefore, prosthetic stiffness probably influences long jump performance and should be subject of future research.

As shown in our previous analysis of the same jumps, ground contact times of the take-off step were not different between non-amputee athletes ($122 \pm 9$ ms) and athletes with BKA ($124 \pm 14$ ms) who used their affected leg as their take-off leg [12]. For non-amputees, Lees et al. [15] define that the duration of the pivot equals the compression phase, from TD to MKF. During this phase, athletes generate 64–70% of the vertical take-off velocity [14,15,19]. Even though our results show different COM velocity profiles between athletes with BKA and non-amputee athletes from TD to TO of the take-off step (figure 2 and table 2), the generation of vertical velocity during the compression phase relative to the vertical velocity at TO was not different between groups (BKA: 59.3%, non-AMP: 60.5%), but slightly lower compared to previous research [14,15,19].

Good long jump performance of non-amputee athletes was related to their ability to tolerate high-impact forces [29] and to increase the height of the COM immediately after TD [30]. By contrast, other studies found that vertical velocity has a downward orientation at TD [13,15,19,20] that lasts for about 5% of the take-off step [19]. The vertical COM movement of the non-amputee athletes from the current study is in agreement with the results of Lees et al. [15], but vertical COM increase during the take-off step was about 7–8 cm lower than reported in other studies [19,20]. Based on the conflicting ideas mentioned above and a previous discussion by Hay et al. [13], an immediate upward movement of the COM at TD may not be a robust predictor of good long jump performance. Additionally, our results show that the vertical COM movement of athletes with BKA is different compared to non-amputee long jumpers presented here or previously [15,19]. Total vertical displacement of the COM was smaller in the athletes with BKA, but the vertical COM downward displacement lasted more than twice as long compared to non-amputee athletes. The latter, in combination with greater shortening of the lower leg during the compression phase and a 20.2% longer relative duration of the compression phase, emphasizes the inability of athletes with BKA to increase leg stiffness in their affected leg [23]. Thus, long jumpers with BKA must alter their movement strategy and predominately rely on energy storage and return in the RSP. By contrast, Luhtanen & Komi [29] argue that good non-amputee long jumpers are able to resist high-impact forces and subsequently 'benefit more from the elastic behavior of the muscles' [29, p. 273] during the take-off step. Therefore, it is important to differentiate between the required take-off strategies for the long jump elicited by non-amputee athletes and athletes with BKA who use their affected leg as their take-off leg. By using their RSP as a spring, the athletes with BKA had a vertical COM downward displacement during the first 42% of ground contact but still benefited from elastic energy storage and return, and thus elicited the same long jump performance. The movement patterns of athletes with BKA are likely to be dictated by the use of a prosthesis. Neither in the present nor in previous studies, have such patterns been reported for non-amputee long jumpers.

Based on COM kinematics during the take-off step, the determinants of long jump performance are different for non-amputees and athletes with BKA. Further research is needed to identify performance limiting factors for the long jump for athletes with BKA (e.g. duration and extent of the downward movement, as this is directly affected by the stiffness of the prosthesis). Knowledge about performance limiting factors may help coaches and biomechanical staff to adapt and enhance the performance analysis for long jumpers with BKA.

## 4.2. Non-sagittal plane COM kinematics

Medio-lateral COM position relative to the take-off foot was different throughout the stance phase between groups. The COM position was about 5 cm medial to the COP during most of the take-off step for athletes with BKA, whereas the non-amputee athletes had their COM 0.6 cm medial to the COP at TD and 1.4 cm lateral to the COP at TO. The results from non-amputees are in line with previous studies, which found that the COM was directly above the ankle at TD and then deviated laterally until TO [19,20]. Medial placement of the foot results in foot eversion during the early phase of ground contact (figure 4) to create a flat support area. Because an RSP is designed to primarily act in the sagittal plane, athletes with BKA cannot evert or invert their RSP, and a medial foot placement would result in a force applied to the lateral edge of the RSP, which would probably compromise

dynamic stability and could result in a failed jump attempt or fall. The difference in prosthetic position of athletes with BKA compared to non-amputee athletes is probably constrained and induced by the RSP's design and rigidity in the frontal plane.

During ground contact of the take-off step, non-amputee athletes had increasing lateral velocity, which resulted in a distinct 2.6 cm lateral COM displacement at TO (table 2). This lateral COM displacement probably results from placing the take-off foot beneath the COM and the resulting lateral ground reaction force (GRF) peak during the early phase of ground contact [12]. Athletes with BKA did not have a lateral GRF peak [12] and, therefore, did not have a lateral COM displacement.

The effect of a lateral take-off velocity on absolute horizontal jump distance can be calculated by using the aerial time calculated in our previous analysis for the same jumps [7]. An aerial time of 0.882 s [7] and a take-off velocity of 0.38 m s$^{-1}$ in the lateral direction, shown by the non-amputee athletes (table 2), would result in a sidewise COM displacement of 0.34 m during the flight phase— neglecting air resistance. Using trigonometry and a theoretical jump distance of 7.26 m, a lateral displacement of 0.34 m would result in a 0.8 cm increase in the absolute linear horizontal jump distance. Therefore, a lateral take-off velocity has a minimal effect on non-amputee athletes' long jump performance.

## 4.3. Sagittal plane joint angles

Hip flexion angles of non-amputee athletes at TD were about 6–9° lower in this study compared to data from previous studies of non-amputee long jumpers, who reached effective distances of 7.45 ± 0.18 m [19], 7.96 ± 0.15 m [20] and 7.79 ± 0.24 m [15], respectively. Furthermore, non-amputee athletes from the present study did not reach peak hip flexion prior to 25 ± 1.7% of the stance phase, and then reached maximum extension of the hip at TO with values similar to those in [19,20] but about 12° greater compared to those in [15]. This discontinuous sagittal plane hip motion contrasts with previous studies that reported continuous hip extension throughout ground contact [19,20]. These differences between studies, especially during the early stance phase of the take-off step, might be due to a shorter average jump distance in our study or due to lower sampling rates (100 or 125 Hz) [15,19,20] and lower cut-off frequencies (8 or 8.3 Hz) [15,19] used in previous studies. When analysing the ground contact phase of the long jump take-off step, which lasts about 0.12 s, low sampling rates and cut-off frequencies underestimate peak values by smoothing the data. This could lead to the neglect of values potentially important for identifying movement strategies. Our study (together with [7] and [12]) is the first to use an optoelectronic, marker-based system for data collection as well as sampling frequencies [25,31] and filtering regimes [24,25] similar to those recently used for sprint running analyses. Compared to non-amputee long jumpers, athletes with BKA have lower peak ground reaction forces in the vertical and posterior directions during the take-off step [7,12], which result in lower sagittal plane peak hip and knee joint moments [12]. Therefore, athletes with BKA had less knee flexion and had continuous hip joint extension compared to non-amputee athletes.

The knee flexion angle of the non-amputee athletes was similar at TD, about 7–13° higher at MKF and about 2–7° lower at TO compared to previous studies of non-amputee long jumpers [15,19,20]. As discussed previously, these differences could be due to different jump distances or different sampling rates and filtering procedures. Hip and knee joint sagittal ROM were lower (both $p = 0.017$) in athletes with BKA (35.6°, 22.5°) compared to non-amputee athletes (60.1°, 44.4°) (figure 4). Apparently, athletes with BKA seek to keep their leg straight during the take-off step of the long jump, most likely to optimize the energy exchange with the prosthesis. A similar strategy of stiffening the affected leg was observed when comparing the take-off step characteristics of athletes with BKA using their unaffected leg versus athletes with BKA using their affected leg for the take-off step [11]. The effectiveness of this prosthetic 'springboard' [11, p. 304] motion probably depends on the prosthetic configuration (e.g. stiffness, height) and its alignment. However, the choice of prosthetic configuration is subjective and not based on systematic research. Future research should determine the effects of different prosthetic configurations on long jump performance in athletes with BKA and further research is needed that addresses the influence of socket fit (interface between the RSP and residual limb) on performance.

## 4.4. Non-sagittal plane joint angles

Frontal plane hip angles for non-amputee athletes are in accordance with values reported in previous studies [19,20]; the hip is abducted at TD and TO. During the first half of the stance phase,

non-amputee athletes were not able to resist hip adduction, which is shown by a slight frontal plane bend of the thigh relative to the pelvis (figure 4). Athletes with BKA, in general, had a similar, but less pronounced pattern for frontal plane hip and knee motion compared to non-amputee athletes. Frontal plane joint motion confirms the importance of muscles relevant for frontal plane stabilization and body weight support [32] in both groups of athletes.

To our knowledge, this is the first study to report transverse plane joint angles during the long jump take-off step. Transverse plane hip and knee angles of non-amputee long jumpers are similar to those reported for sprinting [25]. One study found that the ankle was predominantly externally rotated during ground contact for non-amputee sprinters [25], whereas the ankle joint was internally rotated during the take-off step of non-amputee long jumpers in the present study. The internally rotated ankle may result from the non-amputee long jumpers' ambition to position their foot beneath the COM (figure 3). However, during the compression phase of the take-off step, the ankle of the non-amputees rotated externally from $10.6°$ internal rotation at TD to $2.9°$ internal rotation at MKF (table 3 and figure 4). An external rotation of the foot relative to the shank segment implies an internal rotation of the tibia relative to the foot. Together with the frontal plane foot eversion (figure 4), the non-amputee long jumpers in the present study elicited motions that resemble the tibiocalcaneal coupling previously reported for the stance phases of sprinting [25] and running [33].

## 4.5. Limitations

A limited number of long jumpers with BKA participated in this study; however, these include three out of the four best long jumpers who participated in the 2016 Paralympic Games. Due to the high inter-athlete performance differences in Paralympic long jump, a greater sample size might increase variability and the risk of overlooking movement patterns unique for long jumpers with BKA competing at the highest performance level. Although there are advantages of optoelectrical systems (e.g. high accuracy, no manual digitizing), marker movement on the skin in high-impact movements might influence joint angle calculations. The individual alignment and type of prosthesis used by athletes with BKA probably affect take-off step kinematics. However, all of the RSP models currently used by elite athletes provide the same spring-like function; thus, there are not likely differences in movement strategies between athletes due to different RSP models. In the present analysis, all athletes with BKA used the same type of RSP (Cheetah Xtreme; Össur, Reykjavik, Iceland) with their individual alignment. Future research should investigate the effects of different RSP configurations that optimize fit and performance.

## 5. Conclusion

Long jumpers with BKA positioned their prosthesis more laterally compared to the foot placement beneath the COM used by non-amputee athletes. This strategy may avoid GRF placement on the edge of the prosthesis and, furthermore, may explain the absence of any relevant medio-lateral GRF application [12] and medio-lateral COM movement. Compared to non-amputee athletes, long jumpers with BKA had less sagittal plane ROM in the hip and knee joints during stance but greater shortening of the whole leg and lower leg during the compression phase, which is due to the compression of the prosthesis compared to the biological leg. In general, long jumpers with BKA had a longer compression phase and a prominent downward movement of the COM, indicating a less rigid lever compared to the biological leg of non-amputees. Thus, the redirection of horizontal velocity to vertical velocity in athletes with BKA is different compared to non-amputee athletes. The motion of athletes with BKA does not reflect the strategy of pivoting shown by non-amputee athletes, but resembles the use of a 'springboard' [11, p. 304]. However, these different take-off step techniques lead to the same long jump performance.

In our previous paper [7], we pointed out how non-amputee long jumpers cannot adopt the technique elicited by athletes with BKA, even if they would intend to, due to their limited capacity for storing and returning energy in their biological structures. On the other hand, in the present study, we show that, due to the mechanical constraints of the prosthesis, long jumpers with BKA are unable to actively regulate their lower leg stiffness and have to regulate the stiffness of the whole leg by reducing knee ROM during the take-off step.

In sum, long jumpers with BKA who use their affected leg as their take-off leg cannot adopt the take-off technique elicited by non-amputee long jumpers and vice versa. Thus, our results can be used to enhance

athletic performance of long jumpers with BKA and the design of future prostheses. Additionally, they may inform decision makers during the revision of Olympic and Paralympic regulations.

Ethics. All athletes gave voluntary informed consent to participate in the study. The study design was in line with the declaration of Helsinki and was approved by the ethical committee board of the German Sport University Cologne (approval number: 040/2016).

Data accessibility. Data available from the Dryad Digital Repository: https://doi.org/10.5061/dryad.t7d934c [34].

Authors' contributions. W.P., H.H., A.M.G., S.W., J.F. and R.M. designed and managed the study. Data capturing was executed by J.F., S.W., R.M., H.H., A.M.G. and W.P. Model and parameter calculations were performed by K.H., J.F. and S.W. All authors interpreted the data. J.F. drafted and wrote the manuscript with contribution from A.M.G., S.W., H.H., W.P., R.M. and K.H. All authors gave final approval for publication.

Competing interests. None of the authors has any conflict of interest associated with the study.

Funding. Funding was provided by the Japan Broadcasting Cooperation (NHK). J.F. was funded by a graduate fellowship of the German Sport University Cologne. H.H. was funded by JSPS KAKENHI grant no. 26702027. A.M.G.'s contribution to this project was partially supported by the BADER Consortium, a US Department of Defense Congressionally Directed Medical Research Programs cooperative agreement (W81XWH-11-2-0222).

Acknowledgements. We are thankful to all athletes for participating in this study during an important training period of an Olympic/Paralympic season. We are also grateful to the technical staff of the German Sport University Cologne and the Japan Institute of Sport Sciences for their outstanding support and thorough preparations made for the data collections. We thank Adrian Vincent for the initial proof reading of the manuscript. Furthermore, we thank Stephan Dill, Denis Holzer, Anna Lena Kleesattel, Igor Komnik, Markus Kurz, Markus Peters, Erik Schrödter, Josef Viellehner and Jana Weichsel for their fantastic help during data collection and post-processing.

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
