## [Reviewer comments · Royal Society Open Science]

Review History

RSOS-190107.R0 (Original submission)

Review form: Reviewer 1

Is the manuscript scientifically sound in its present form?

Yes

Are the interpretations and conclusions justified by the results?

Yes

Is the language acceptable?

Yes

Is it clear how to access all supporting data?

No

Do you have any ethical concerns with this paper?

No

Have you any concerns about statistical analyses in this paper?

I do not feel qualified to assess the statistics

Recommendation?

Accept with minor revision (please list in comments)

Comments to the Author(s)

The authors present a study in which they compared the techniques of long jumpers with and without a unilateral transtibial amputation. They focused on studying three amputees that have competed in the Paralympics and jump using the prosthesis as the take-off leg. These athletes were then compared to seven non-amputee athletes. They did this by recording motion capture and force plate data during the take-off step and calculating full body kinematics using a mathematical rigid body model. From their research, they find that, while both amputees and non-amputees displayed similar performance, they used different strategies for producing their greatest distances. Amputees experienced a longer compression phase at the start of the take-off step, kept their intact joints much stiffer, and had a lower COP distance from their center of mass. This paper is well written and well reasoned and I such I believe it is worthy of publication after a few minor edits and clarifications.

Revisions (relevant line numbers are shown):

115-120: Where both amputee and non-amputee subjects present at both locations (GSU and JISS)? This seems necessary to mention as there were differences in data collection at the two sites. If the two groups had data collected at separate locations, could that affect the results?

150-152: What are the 16 segments of the model? A citation that includes a detailed description of the model and/or labeled segments in the figure would be helpful. If there is no previous study to cite, the authors may need to write a supplementary section describing the model. I would also recommend they post the data and code for the study.

155: The modeled prosthesis joint should be labeled in figure 1. Describing the location becomes much clearer with a picture.

177: How does landing technique influence the information gathered at take-off?

187: The study cited here gives no extra information on the statistical technique used. From what I can gather from the p values reported throughout the Results section, the null hypothesis always assumes the data sets are the same. While I am admittedly not and statistical expert, the stats throughout the next section were hard for me to follow.

Results section general comment: I found the statistics to be difficult to understand. Why was the decision made to state the p value in the middle of the comparison, e.g. $A > (p = 0.05) B$, rather than at the end, e.g. $A > B (p = 0.05)$?

223-226: The authors suddenly begin using "significantly different" to compare groups. This is not consistent with previous or future paragraphs. I recommend removing "significantly" or adding it where appropriate to other comparisons.

247-258: Results here lack stats as opposed to all other measures. If there is a reason for the omission, the authors should be explicit.

292: "...indicating that their leg is not as rigid..." This statement can be a bit confusing depending on if the reader includes the prosthesis as a part of the leg or not as previously you state "...they resist knee flexion more..." The authors should consider being explicit throughout this section when speaking about the "whole leg" or the "lower leg."

306: If the ability to tolerate high impact force is a predictor of performance for a non-amputee, wouldn't a stiffer spring on an amputee (as suggested earlier) be counter-productive?

383-385: Does a lower frequency in motion capture data affect the smoothness of the data as well? If so, wouldn't the two different capture frequencies at GSU and JISS affect the results?

401-404: This sentence is hard to understand. Is this statement saying that amputees who jump off their unaffected limb do not stiffen like those who jump off their affected limb?

Figures and Tables:

Most figures report distance in meters (except for two plots in Figure 3) while Table 2 reports in cm. I realize this is done for the most part based on scale, but I'd recommend at least changing Figure 3 for consistency in units.

Review form: Reviewer 2

Is the manuscript scientifically sound in its present form?

Yes

Are the interpretations and conclusions justified by the results?

Yes

Is the language acceptable?

Yes

Is it clear how to access all supporting data?

No

Do you have any ethical concerns with this paper?

No

Have you any concerns about statistical analyses in this paper?

No

Recommendation?

Accept with minor revision (please list in comments)

Comments to the Author(s)

General comments

Overall this is an interesting article with some novel results. A few methodological issues need to be clarified. I feel that the main improvements would come from shortening the results section and improving the discussion of the mechanical processes underpinning the jump takeoff in able bodied and amputee athletes.

Specific comments

Line 62 - insert 'Male' between 'for' and 'athletes'

Lines 126-127 - Presumably these anthropometric measurements were used to create a geometric model of the body for inertia parameter calculation and CoM location, could you specify which one you used?

Lines 156-157 - Did you establish how well the rigid body model fitted the prosthesis - e.g. how close were the markers to the model joint, assuming invariant segment lengths and endpoints?

Lines 163-165 - How were COM position and velocity calculated? From the inertia model throughout, or from initial position and velocity estimates from the inertia model at touchdown and integration of force data? The latter would seem a more accurate estimate, or at least useful to establish whether the two methods agree.

Lines 193-194 - The jump distances were essentially equivalent between groups based on projectile motion, however horizontal and vertical velocities at takeoff were higher in non-amputees (Table 2), amputee athletes would require a substantially higher takeoff position to result in similar jump distances, can you explain this?

Line 193-272 - The results section is long and repeats a lot of the information from the tables and figures. I think this should be briefer and try to restrict the text to highlighting the most extreme or noteworthy findings, rather than reporting most of the data which is already presented in tables and figures. These can be covered in the discussion. Ideally the tables would have some indication of different p-value levels for comparisons between groups.

lines 289-295 - The implication here seems to be that a more rigid prosthesis would be beneficial for a jump takeoff, due to a more effective pivot, however a stiffer prosthesis may result in more flexion of the knee which inevitably results in energy loss due to the force velocity relationship of muscle (i.e. they produce more force - and hence lose more energy - eccentrically than concentrically). A prosthesis which returns nearly all the energy it stores is more effective than an intact limb, even if it results in more overall shortening of the limb, and especially if it facilitates less flexion of the knee joint. If an intact limb athlete took off from a springboard the effect would likely be similar, the longer compression phase would facilitate lower GRFs and consequently less energy loss due to joint flexion, and a more efficient velocity conversion when the energy was returned. Stiffening a prosthesis to mimic intact limb leg stiffness would be like choosing to stiffen the springboard to the same level as the ground, which would likely result in poorer performance due to greater energy losses.

Lines 321-325 - I would contend that movement strategies are the similar between amputees and non-amputees, both are trying to resist knee flexion and therefore energy loss, and use hip extension to add energy to the CoM, but the prosthesis facilitates this more effectively due to reduced GRFs.

Lines 398-401 - As previously mentioned, shortening of the leg due to prosthesis deformation is unlikely to be deleterious, so should not need compensating for.

Lines 464-465 - Although long jump performance was similar between groups (although this needs clarifying, see question re lines 193-194 above), approach velocity was lower in amputee athletes so it is likely their takeoff technique is more effective and any limitations of the prosthesis may have manifested themselves in the approach run.

Decision letter (RSOS-190107.R0)

07-Mar-2019

Dear Mr Funken

On behalf of the Editors, I am pleased to inform you that your Manuscript RSOS-190107 entitled "Long jumpers with and without a transtibial amputation have different 3D center of mass and joint take-off step kinematics" has been accepted for publication in Royal Society Open Science subject to minor revision in accordance with the referee suggestions. Please find the referees' comments at the end of this email.

The reviewers and handling editors have recommended publication, but also suggest some minor revisions to your manuscript. Therefore, I invite you to respond to the comments and revise your manuscript.

- Ethics statement

- Data accessibility

<http://datadryad.org/submit?journalID=RSOS&manu=RSOS-190107>

- Competing interests

- Authors' contributions

- Acknowledgements

- Funding statement

Because the schedule for publication is very tight, it is a condition of publication that you submit the revised version of your manuscript before 16-Mar-2019. Please note that the revision deadline will expire at 00.00am on this date. If you do not think you will be able to meet this date please let me know immediately.

- 1) A text file of the manuscript (tex, txt, rtf, docx or doc), references, tables (including captions) and figure captions. Do not upload a PDF as your "Main Document";
- 2) A separate electronic file of each figure (EPS or print-quality PDF preferred (either format should be produced directly from original creation package), or original software format);
- 3) Included a 100 word media summary of your paper when requested at submission. Please ensure you have entered correct contact details (email, institution and telephone) in your user account;
- 4) Included the raw data to support the claims made in your paper. You can either include your data as electronic supplementary material or upload to a repository and include the relevant doi

within your manuscript. Make sure it is clear in your data accessibility statement how the data can be accessed;

5) All supplementary materials accompanying an accepted article will be treated as in their final form. Note that the Royal Society will neither edit nor typeset supplementary material and it will be hosted as provided. Please ensure that the supplementary material includes the paper details where possible (authors, article title, journal name).

on behalf of Dr Manoj Srinivasan (Associate Editor) and Kevin Padian (Subject Editor)
openscience@royalsociety.org

Reviewer comments to Author:
Reviewer: 1

Comments to the Author(s)

The authors present a study in which they compared the techniques of long jumpers with and without a unilateral transtibial amputation. They focused on studying three amputees that have competed in the Paralympics and jump using the prosthesis as the take-off leg. These athletes were then compared to seven non-amputee athletes. They did this by recording motion capture and force plate data during the take-off step and calculating full body kinematics using a mathematical rigid body model. From their research, they find that, while both amputees and non-amputees displayed similar performance, they used different strategies for producing their greatest distances. Amputees experienced a longer compression phase at the start of the take-off step, kept their intact joints much stiffer, and had a lower COP distance from their center of mass.

This paper is well written and well reasoned and I such I believe it is worthy of publication after a few minor edits and clarifications.

Revisions (relevant line numbers are shown):

115-120: Where both amputee and non-amputee subjects present at both locations (GSU and JISS)? This seems necessary to mention as there were differences in data collection at the two sites. If the two groups had data collected at separate locations, could that affect the results?

150-152: What are the 16 segments of the model? A citation that includes a detailed description of the model and/or labeled segments in the figure would be helpful. If there is no previous study to cite, the authors may need to write a supplementary section describing the model. I would also recommend they post the data and code for the study.

155: The modeled prosthesis joint should be labeled in figure 1. Describing the location becomes much clearer with a picture.

177: How does landing technique influence the information gathered at take-off?

187: The study cited here gives no extra information on the statistical technique used. From what I can gather from the p values reported throughout the Results section, the null hypothesis always assumes the data sets are the same. While I am admittedly not and statistical expert, the stats throughout the next section were hard for me to follow.

Results section general comment: I found the statistics to be difficult to understand. Why was the decision made to state the p value in the middle of the comparison, e.g. $A > (p = 0.05) B$, rather than at the end, e.g. $A > B (p=0.05)$?

223-226: The authors suddenly begin using "significantly different" to compare groups. This is not consistent with previous or future paragraphs. I recommend removing "significantly" or adding it where appropriate to other comparisons.

247-258: Results here lack stats as opposed to all other measures. If there is a reason for the omission, the authors should be explicit.

292: "...indicating that their leg is not as rigid..." This statement can be a bit confusing depending on if the reader includes the prosthesis as a part of the leg or not as previously you state "...they resist knee flexion more..." The authors should consider being explicit throughout this section when speaking about the "whole leg" or the "lower leg."

306: If the ability to tolerate high impact force is a predictor of performance for a non-amputee, wouldn't a stiffer spring on an amputee (as suggested earlier) be counter-productive?

383-385: Does a lower frequency in motion capture data affect the smoothness of the data as well? If so, wouldn't the two different capture frequencies at GSU and JISS affect the results?

401-404: This sentence is hard to understand. Is this statement saying that amputees who jump off their unaffected limb do not stiffen like those who jump off their affected limb?

Figures and Tables:

Most figures report distance in meters (except for two plots in Figure 3) while Table 2 reports in cm. I realize this is done for the most part based on scale, but I'd recommend at least changing Figure 3 for consistency in units.

Reviewer: 2

Comments to the Author(s)

General comments

Overall this is an interesting article with some novel results. A few methodological issues need to be clarified. I feel that the main improvements would come from shortening the results section and improving the discussion of the mechanical processes underpinning the jump takeoff in able bodied and amputee athletes.

Specific comments

Line 62 - insert 'Male' between 'for' and 'athletes'

Lines 126-127 - Presumably these anthropometric measurements were used to create a geometric model of the body for inertia parameter calculation and CoM location, could you specify which one you used?

Lines 156-157 - Did you establish how well the rigid body model fitted the prosthesis - e.g. how close were the markers to the model joint, assuming invariant segment lengths and endpoints?

Lines 163-165 - How were COM position and velocity calculated? From the inertia model throughout, or from initial position and velocity estimates from the inertia model at touchdown and integration of force data? The latter would seem a more accurate estimate, or at least useful to establish whether the two methods agree.

Lines 193-194 - The jump distances were essentially equivalent between groups based on projectile motion, however horizontal and vertical velocities at takeoff were higher in non-amputees (Table 2), amputee athletes would require a substantially higher takeoff position to result in similar jump distances, can you explain this?

Line 193-272 - The results section is long and repeats a lot of the information from the tables and figures. I think this should be briefer and try to restrict the text to highlighting the most extreme or noteworthy findings, rather than reporting most of the data which is already presented in tables and figures. These can be covered in the discussion. Ideally the tables would have some indication of different p-value levels for comparisons between groups.

lines 289-295 - The implication here seems to be that a more rigid prosthesis would be beneficial for a jump takeoff, due to a more effective pivot, however a stiffer prosthesis may result in more flexion of the knee which inevitably results in energy loss due to the force velocity relationship of muscle (i.e. they produce more force - and hence lose more energy - eccentrically than concentrically). A prosthesis which returns nearly all the energy it stores is more effective than an intact limb, even if it results in more overall shortening of the limb, and especially if it facilitates less flexion of the knee joint. If an intact limb athlete took off from a springboard the effect would likely be similar, the longer compression phase would facilitate lower GRFs and consequently less energy loss due to joint flexion, and a more efficient velocity conversion when the energy was returned. Stiffening a prosthesis to mimic intact limb leg stiffness would be like choosing to stiffen the springboard to the same level as the ground, which would likely result in poorer performance due to greater energy losses.

Lines 321-325 - I would contend that movement strategies are the similar between amputees and non-amputees, both are trying to resist knee flexion and therefore energy loss, and use hip

extension to add energy to the CoM, but the prosthesis facilitates this more effectively due to reduced GRFs.

Lines 398-401 - As previously mentioned, shortening of the leg due to prosthesis deformation is unlikely to be deleterious, so should not need compensating for.

Lines 464-465 - Although long jump performance was similar between groups (although this needs clarifying, see question re lines 193-194 above), approach velocity was lower in amputee athletes so it is likely their takeoff technique is more effective and any limitations of the prosthesis may have manifested themselves in the approach run.

Author's Response to Decision Letter for (RSOS-190107.R0)

See Appendix A.

Decision letter (RSOS-190107.R1)

27-Mar-2019

Dear Mr Funken,

I am pleased to inform you that your manuscript entitled "Long jumpers with and without a transtibial amputation have different 3D center of mass and joint take-off step kinematics" is now accepted for publication in Royal Society Open Science.

on behalf of Dr Manoj Srinivasan (Associate Editor) and Professor Kevin Padian (Subject Editor)
openscience@royalsociety.org

Appendix A

To the associate editor of *Royal Society Open Science*

Dear Dr Manoj Srinivasan,

Dear Mr. Dunn,

we would like to thank you for the response regarding the above manuscript and the opportunity to revise and address the comments made by the two expert reviewers. Also, we would like to thank the expert reviewers for taking their time to read and comment on our manuscript; the comments were clearly the result of a thorough and very insightful review. We feel that the revisions made have added to the strength of the manuscript.

The specific reviewer comments have been addressed below. An amended version of the manuscript has been re-submitted.

We also went through the whole document, improved the quality of the language, and corrected some minor imprecisions and typing errors.

Reviewer comments to Author:

Reviewer: 1

Comments to the Author(s)

The authors present a study in which they compared the techniques of long jumpers with and without a unilateral transtibial amputation. They focused on studying three amputees that have competed in the Paralympics and jump using the prosthesis as the take-off leg. These athletes were then compared to seven non-amputee athletes. They did this by recording motion capture and force plate data during the take-off step and calculating full body kinematics using a mathematical rigid body model. From their research, they find that, while both amputees and non-amputees displayed similar performance, they used different strategies for producing their greatest distances. Amputees experienced a longer compression phase at the start of the take-off step, kept their intact joints much stiffer, and had a lower COP distance from their center of mass. This paper is well written and well reasoned and I such I believe it is worthy of publication after a few minor edits and clarifications.

Revisions (relevant line numbers are shown):

115-120: Where both amputee and non-amputee subjects present at both locations (GSU and JISS)? This seems necessary to mention as there were differences in data collection at the two sites. If the two groups had data collected at separate locations, could that affect the results?

We understand your concerns. However, there was no group-location assignment. Eight athletes (two with and six without BKA) were tested at the German Sport University Cologne (GSU), two athletes (one with and one without BKA) were tested at the Japan Institute of Sport Sciences. We agree that the location of data collection might affect the results. To minimize the effect, two researchers from GSU (first and second author) traveled to Japan for data collection and to ensure standardization of procedures. In both locations, a Vicon system and Kistler forces plates were used.

We added the required information and rewrote as follows:

We changed:

“Data collection was conducted at the German Sport University Cologne (GSU) and the Japan Institute of Sport Sciences (JISS).”

To:

“Data collection was conducted at the German Sport University Cologne (GSU, six athletes with and two without BKA) and the Japan Institute of Sport Sciences (JISS, one athlete with and one without BKA).”

150-152: What are the 16 segments of the model? A citation that includes a detailed description of the model and/or labeled segments in the figure would be helpful. If there is no previous study to cite, the authors may need to write a supplementary section describing the model. I would also recommend they post the data and code for the study.

We are thankful to the reviewer for pointing out this lack of information. We added the required information and rewrote as follows:

We changed:

“A mathematical rigid full body model (Dynamicus, Alaska, Institute of Mechatronics, Chemnitz, Germany) consisting of sixteen segments was used for kinematic calculations for the non-amputee athletes.”

To:

“A mathematical rigid full body model (Dynamicus, Alaska, Institute of Mechatronics, Chemnitz, Germany) consisting of sixteen main segments was used for kinematic calculations for the non-amputee athletes. The segments comprised: head, trunk, as well as left and right: upper arm, lower arm, hand, thigh, shank, rear foot, forefoot. The trunk consisted of several sub-segments.”

As stated at the end of the method section, additional, for this study relevant, details on the mathematical model (e.g., joint center definition) and calculation of jump distance are in our previous paper Willwacher et al. (2017).

Unfortunately, we cannot upload marker coordinate data or the model used. This is due to our small sample size and that we cannot guarantee the anonymity of subject-specific data. As we said, all models for the athletes with BKA are adapted individually. This also means that it includes personal measurements. E.g. residual shank length etc.

However, we have uploaded all model output files which were used for parameter calculation.

155: The modeled prosthesis joint should be labeled in figure 1. Describing the location becomes much clearer with a picture.

Thank you for this suggestion. We agree and added a label to Figure 1, highlighting the position of the prosthetic joint. We also changed the figure caption accordingly.

177: How does landing technique influence the information gathered at take-off?

As mentioned in the methods section we eliminated the influence of landing technique on long jump performance by calculating the theoretical jump distance.

We do not expect that the landing technique will influence the information gathered at take-off.

On the other side, the technique used during the flight phase (e.g. run or hang) could influence landing technique. In the current study we did not capture the actual landing, but only a limited range of 2-3 meters behind the take-off board. Therefore, stating influences of the technique used during the flight phase on landing technique would be pure speculation and would be beyond the scope of this paper. However, we agree that this might be important for athletes and coaches and should be subject of further research. Personally, we did not observe differences in landing or flight technique between athletes with and without BKA. However, inter-individual differences appear in both groups.

187: The study cited here gives no extra information on the statistical technique used. From what I can gather from the p values reported throughout the Results section, the null hypothesis always assumes the data sets are the same. While I am admittedly not and statistical expert, the stats throughout the next section were hard for me to follow.

It is correct that the null hypothesis assumes no difference between groups. We cited our previous publication (Willwacher et al., 2017) because we used the same statistical procedure, a Wilcoxon test and 10% error margin.

We added the required information as follows:

“We assumed that the null hypothesis was indicated as no difference between groups.”

Results section general comment: I found the statistics to be difficult to understand. Why was the decision made to state the p value in the middle of the comparison, e.g. $A > (p = 0.05) B$, rather than at the end, e.g. $A > B (p=0.05)$?

Thank you for this comment. We understand the issue. We tried to move the position of the p-value to the end of the sentence or statement wherever possible without causing confusion.

223-226: The authors suddenly begin using “significantly different” to compare groups. This is not consistent with previous or future paragraphs. I recommend removing “significantly” or adding it where appropriate to other comparisons.

We agree. We removed „significantly“ for consistency.

247-258: Results here lack stats as opposed to all other measures. If there is a reason for the omission, the authors should be explicit.

We thank the reviewer for highlighting this inconsistency. We added p-values.

292: “...indicating that their leg is not as rigid...” This statement can be a bit confusing depending on if the reader includes the prosthesis as a part of the leg or not as previously you state “...they resist knee flexion more...” The authors should consider being explicit throughout this section when speaking about the “whole leg” or the “lower leg.”

Thank you for pointing this out. We agree and rephrased to make it clearer.

We changed:

Resisting knee flexion would allow athletes to keep their leg stiff, providing a stable lever.

To:

Resisting knee flexion would allow non-amputee athletes to keep their leg stiff, providing a stable lever.

This underlines that non-amputees can stiffen their whole leg by resisting knee flexion.

306: If the ability to tolerate high impact force is a predictor of performance for a non-amputee, wouldn't a stiffer spring on an amputee (as suggested earlier) be counter-productive?

We did not intend to implicate that the prosthesis should be stiffer and have rephrased the sentence to make it clearer.

We changed:

“Because leg stiffness is dictated by RSP stiffness during running [28], use of a stiffer prosthesis by an athlete with BKA could increase leg stiffness during the long jump take-off step, but may also impair the elastic energy storage and return within the leg.”

To:

“Because leg stiffness is dictated by RSP stiffness during running [28], use of a stiffer prosthesis by an athlete with BKA could increase leg stiffness during the long jump take-off step. However, a stiffer prosthesis might impair the elastic energy storage and return within the leg and result in higher impact forces leading to increased knee flexion. Therefore, prosthetic stiffness likely influences long jump performance and should be subject of future research.”

383-385: Does a lower frequency in motion capture data affect the smoothness of the data as well? If so, wouldn't the two different capture frequencies at GSU and JISS affect the results?

With a capturing frequency of 250Hz, we chose a sampling frequency which guarantees to cover the frequency content of such a highly dynamic motion as the take-off step in the long jump. Similar frequencies were used in current research on sprint running (Alt et al 2015, Judson et al 2018). We thus can not only guarantee to keep all kinematic information but we were also able to use a low pass filter with a cut-off frequency of 50 Hz. With 500 Hz we cover the same content of information and because of an adequate and consisting filtering regime, we don't expect any relevant differences.

401-404: This sentence is hard to understand. Is this statement saying that amputees who jump off their unaffected limb do not stiffen like those who jump off their affected limb?

Yes, Nolan et al. 2012 showed that athletes with BKA who take-off from their affected leg have a stiffer knee compared to athletes with BKA who take off from their unaffected leg. This comparison is similar to our comparison between athletes with BKA who take-off from their affected leg and non-amputee athletes.

Figures and Tables:

Most figures report distance in meters (except for two plots in Figure 3) while Table 2 reports in cm. I realize this is done for the most part based on scale, but I'd recommend at least changing Figure 3 for consistency in units.

We agree and now all figures display distances in cm to be consistent with Table 2. We also slightly changed the scaling of two sub-figures in Figure 3 and corrected minor imprecisions in Table 2.

Reviewer: 2

Comments to the Author(s)

General comments

Overall this is an interesting article with some novel results. A few methodological issues need to be clarified. I feel that the main improvements would come from shortening the results section and improving the discussion of the mechanical processes underpinning the jump takeoff in able bodied and amputee athletes.

Specific comments

Line 62 - insert 'Male' between 'for' and 'athletes'

We added „male“ accordingly

Lines 126-127 - Presumably these anthropometric measurements were used to create a geometric model of the body for inertia parameter calculation and CoM location, could you specify which one you used?

The anthropometric data were used to individually adapt the „Dynamicus“ model (Alaska, Institute of Mechatronics, Chemnitz, Germany)

The Dynamicus model provides limited information about the underlying inertial and CoM calculations. However, calculations are mainly based on the following references:

Fluegel, Bernd, Holle Greil, und Karl Sommer. Anthropologischer Atlas: Grundlagen und Daten: Alters- und Geschlechtsvariabilität des Menschen. Frankfurt/Main: Wötzel, 1986. ISBN: 3-925831-00-2

Sziorski, W. M., A. S. Aruin, und W. N. Selujanow. Biomechanik des menschlichen Bewegungsapparates. Berlin, Sportverlag, 1984.

Winter, David A. Biomechanics and motor control of human movement. 4th ed. Hoboken, N.J: Wiley, 2009.

Lines 156-157 - Did you establish how well the rigid body model fitted the prosthesis - e.g. how close were the markers to the model joint, assuming invariant segment lengths and endpoints?

The markers defined the position of the model joint. Thus, agreement of the marker position should be the same for every joint.

Lines 163-165 - How were COM position and velocity calculated? From the inertia model throughout, or from initial position and velocity estimates from the inertia model at touchdown and integration of force data? The latter would seem a more accurate estimate, or at least useful to establish whether the two methods agree.

COM position was calculated from the Dynamicus model. Velocity then was calculated by differentiation of the COM position using a three-point method to avoid shortening the data set by one frame.

We added:

„COM velocity was calculated by mathematical differentiation of the COM position throughout the entire stance phase.“

Lines 193-194 - The jump distances were essentially equivalent between groups based on projectile motion, however horizontal and vertical velocities at takeoff were higher in non-amputees (Table 2), amputee athletes would require a substantially higher takeoff position to result in similar jump distances, can you explain this?

Take-off velocities in anterior-posterior and vertical directions were not significantly different between groups ($p=0.667$ and $p=1.000$, respectively). Therefore, athletes had the same average jump distances. As stated in the caption of Table 2, significant differences are indicated by bold values. Also, vertical COM position at TO was not different between groups ($p=0.833$).

Line 193-272 - The results section is long and repeats a lot of the information from the tables and figures. I think this should be briefer and try to restrict the text to highlighting the most extreme or noteworthy findings, rather than reporting most of the data which is already presented in tables and figures. These can be covered in the discussion. Ideally the tables would have some indication of different p-value levels for comparisons between groups.

We agree that a briefer results section would be easier to read. However, most of the values presented in the results section are p-values and percentage differences. These data are not presented in the tables. Further, the tables present an extensive amount of data. Thus, we split the information between the text and tables.

We carefully went through the result section and were able to shorten a few sentences without losing information.

We changed:

“Athletes with BKA reached their maximum knee flexion (MKF) at $52.7 \pm 2.5\%$ of the stance phase during the take-off step, which was later ($p=0.017$) compared to the non-amputee athletes, who reached MKF at $43.8 \pm 4.9\%$ of the stance phase of the take-off step.”

To:

“Athletes with BKA reached maximum knee flexion (MKF) in their take-off leg later during the take-off step compared to the non-amputee athletes (nonAMP: $43.8 \pm 4.9\%$, BKA: $52.7 \pm 2.5\%$, $p=0.017$).”

We changed

For the first $42.2 \pm 5.8\%$ of the stance phase the vertical COM position of athletes with BKA was below their vertical COM position at the instant of TD (Figure 2). This was a longer relative time duration ($p=0.017$) compared with non-amputees (first $17.2 \pm 3.9\%$ of the stance phase).

To:

In athletes with BKA the vertical COM position was below their vertical COM position at TD for a longer relative duration of ground contact compared with non-amputee athletes (nonAMP: $17.2 \pm 3.9\%$, BKA: $42.2 \pm 5.8\%$, $p=0.017$).

We changed

“At the instance of MKF, the COM of both groups of athletes was posterior relative to the COP, but the COM of athletes with BKA was ~13 cm closer ($p=0.017$) to the COP compared to the non-amputee athletes.”

To

“At MKF, both groups had their COM posterior relative to their COP, but athletes with BKA had their COM ~13 cm closer to their COP compared to non-amputees ($p=0.017$).

lines 289-295 - The implication here seems to be that a more rigid prosthesis would be beneficial for a jump takeoff, due to a more effective pivot, however a stiffer prosthesis may result in more flexion of the knee which inevitably results in energy loss due to the force velocity relationship of muscle (i.e. they produce more force - and hence lose more energy - eccentrically than concentrically). A prosthesis which returns nearly all the energy it stores is more effective than an intact limb, even if it results in more overall shortening of the limb, and especially if it facilitates less flexion of the knee joint. If an intact limb athlete took off from a springboard the effect would likely be similar, the longer compression phase would facilitate lower GRFs and consequently less energy loss due to joint flexion, and a more efficient velocity conversion when the energy was returned. Stiffening a prosthesis to mimic intact limb leg stiffness would be like choosing to stiffen the springboard to the same level as the ground, which would likely result in poorer performance due to greater energy losses.

We did not intend to implicate that the prosthesis should be stiffer and have rephrased the sentence to make it clearer.

We changed:

“Because leg stiffness is dictated by RSP stiffness during running [28], use of a stiffer prosthesis by an athlete with BKA could increase leg stiffness during the long jump take-off step, but may also impair the elastic energy storage and return within the leg.”

To:

“Because leg stiffness is dictated by RSP stiffness during running [28], use of a stiffer prosthesis by an athlete with BKA could increase leg stiffness during the long jump take-off step. However, a stiffer prosthesis might impair the elastic energy storage and return within the leg and result in higher impact forces leading to increased knee flexion. Therefore, prosthetic stiffness likely influences long jump performance and should be subject of future research.”

Lines 321-325 - I would contend that movement strategies are the similar between amputees and non-amputees, both are trying to resist knee flexion and therefore energy loss, and use hip extension to add energy to the CoM, but the prosthesis facilitates this more effectively due to reduced GRFs.

Thank you for this suggestion. We only partly agree here. Even though it would be beneficial for both athletes to keep the knee stiff, only the use of a prosthesis enables the athlete to do so. Knowing this, athletes with BKA alter their strategy and rely on energy storage and return in the prosthesis.

The question of effectiveness (advantage/disadvantage) was stressed in our previous paper (Willwacher et al., 2017 entitled: „Elite long jumpers with below the knee prostheses approach the board slower, but take-off more effectively than non-amputee athletes“) and therefore is not discussed again in the current paper.

However, based on your suggestion we rephrased a sentence in the conclusion section.

We changed:

On the other hand, in the present study we show that long jumpers with BKA are unable to actively control their lower leg or regulate the stiffness of their leg during the take-off step due to the mechanical constraints of the prosthesis.

To:

“On the other hand, in the present study we show that, due to the mechanical constraints of the prosthesis, long jumpers with BKA are unable to actively regulate their lower leg stiffness and have to regulate the stiffness of the whole leg by reducing knee ROM during the take-off step.”

Lines 398-401 - As previously mentioned, shortening of the leg due to prosthesis deformation is unlikely to be deleterious, so should not need compensating for.

We understand your concerns and deleted this sentence.

Lines 464-465 - Although long jump performance was similar between groups (although this needs clarifying, see question re lines 193-194 above), approach velocity was lower in amputee athletes so it is likely their takeoff technique is more effective and any limitations of the prosthesis may have manifested themselves in the approach run.

Thank you for this comment. Please see the previous answer for the first part of your comment (jump distances). We agree with the second part of your comment. In our previous paper (Willwacher et al., 2017 entitled: „Elite long jumpers with below the knee prostheses approach the board slower, but take-off more effectively than non-amputee athletes“) we highlighted that long jumpers with BKA approach the board slower but take off more effectively. As discussed before, we did not stress this in the current paper again.